# Application of Artificial Intelligence (AI) in a Cephalometric Analysis: A Narrative Review

**DOI:** 10.3390/diagnostics13162640

**Published:** 2023-08-10

**Authors:** Michał Kiełczykowski, Konrad Kamiński, Konrad Perkowski, Małgorzata Zadurska, Ewa Czochrowska

**Affiliations:** Department of Orthodontics, Medical University in Warsaw, 02-097 Warsaw, Poland; kielczykowski.michal@gmail.com (M.K.); konradkaminski90@hotmail.com (K.K.); konrad.perkowski@wum.edu.pl (K.P.); malgorzata.zadurska@wum.edu.pl (M.Z.)

**Keywords:** artificial intelligence, cephalometric analysis, convolutional neural networks, orthodontics

## Abstract

In recent years, the application of artificial intelligence (AI) has become more and more widespread in medicine and dentistry. It may contribute to improved quality of health care as diagnostic methods are getting more accurate and diagnostic errors are rarer in daily medical practice. The aim of this paper was to present data from the literature on the effectiveness of AI in orthodontic diagnostics based on the analysis of lateral cephalometric radiographs. A review of the literature from 2009 to 2023 has been performed using PubMed, Medline, Scopus and Dentistry & Oral Sciences Source databases. The accuracy of determining cephalometric landmarks using widely available commercial AI-based software and advanced AI algorithms was presented and discussed. Most AI algorithms used for the automated positioning of landmarks on cephalometric radiographs had relatively high accuracy. At the same time, the effectiveness of using AI in cephalometry varies depending on the algorithm or the application type, which has to be accounted for during the interpretation of the results. In conclusion, artificial intelligence is a promising tool that facilitates the identification of cephalometric landmarks in everyday clinical practice, may support orthodontic treatment planning for less experienced clinicians and shorten radiological examination in orthodontics. In the future, AI algorithms used for the automated localisation of cephalometric landmarks may be more accurate than manual analysis.

## 1. Introduction

Artificial intelligence (AI) is the ability of a machine to imitate logical human behaviour, including complex activities [1]. For the first time, this term was introduced by John McCarthy during a conference at Dartmouth College in 1956 [2]. There are many forms of AI, most notably machine learning (ML), artificial neural networks (ANNs), convolutional neural networks (CNN) and deep learning (DL) [3]. Artificial intelligence is used on a daily basis in the internet search engines (Google) and online private intelligent assistants (Siri), also rapidly evolving in other areas, including medicine. It may contribute to the improved quality of health care due to the increased quality of diagnostic methods and the elimination of diagnostic errors in daily medical practice [4]. In medicine, it is primarily used in radiological diagnosis of neoplastic lesions and in assessing histological specimens regarding the advancement of pathological processes. In gastroenterology, it may assist in detecting and monitoring colon polyps and preventing intestinal cancers; in cardiology, it may assist in the interpretation of ECG results [5]. Medical radiology offers a wide range of AI applications as it relies on digitally coded images that can be easily converted into a computer language [6]. Also, in many areas of dentistry, interest in the use of artificial intelligence has considerably increased in recent years [7]. AI algorithms can be useful in the diagnosis of dental caries, periapical or periodontal diseases, classification of maxillofacial cysts or tumours and localisation of cephalometric landmarks [8].

Analysis of lateral cephalometric radiographs is a method widely used in orthodontic diagnosis and treatment planning. It allows for assessing skeletal relations of the maxilla, mandible and cranial base in the sagittal and vertical dimensions as well as dental relations of the upper and lower teeth to the skeletal bases. It is also used to predict the growth direction in children and adolescents and to evaluate the results of orthodontic treatment. Cephalometric analysis is valuable when planning orthognathic surgery to correct skeletal maloclussions in adults [9]. At present, it is used to identify cephalometric points via their digitalisation on the computer screen utilising software for digital cephalometric analysis. In recent years, AI was employed to perform cephalometric analysis, which is supposed to relieve clinicians’ work and save time. Applications that use AI-based image analysis are becoming more common and available to clinicians.

“Deep learning” utilises convolutional neural networks (CNNs) and is the most frequently used algorithm for AI image analysis. The concept of deep learning is based on the following: the algorithm is subjected to pairs of data and corresponding data labels, which in the case of computer vision, will correspond to the images and definitions of the described parameters. In the learning phase, pairs of data and data labels are repeatedly shown to the algorithm, so it becomes optimised to minimise errors in the predicted models. A well-trained algorithm is able to evaluate the structure of the data input and its association with a given label and ultimately is able to predict data labelling on new data/images [10]. A major limitation in training and testing DL models for cephalometric analysis of radiographs is the complexity of data labelling. There is not one true localisation of a specific anthropometric point which would serve as a gold standard. Usually, many specialists manually mark a specific point, and the unification of measurements serves to determine the right label for DL algorithm learning.

The aim of this paper was to present data from the literature on the effectiveness of AI in orthodontic diagnostics based on the analysis of lateral cephalometric radiographs.

## 2. Methodology

PubMed, Medline, Scopus and Dentistry & Oral Sciences Source databases were searched for publications from years 2009 to 2023 to review information on the accuracy of AI in orthodontic diagnostics based on analysis of lateral cephalometric radiographs. The following terms were used: artificial intelligence and cephalometric analysis. After removing duplicates, twenty-three articles were selected based on inclusion criteria.

## 3. Results

Several authors have studied the accuracy of detecting key landmarks using artificial intelligence in cephalometric analysis. Table 1 presents the summarised studies on the application of AI in cephalometric analysis. In total, 23 articles were included based on both AI algorithms designed by their authors for the purpose of a specific study [11,12,13,14,15,16,17,18,19,20,21,22,23,24,25] and web-based software available on search engines and mobile applications [26,27,28,29,30,31,32,33]. The studies focused on comparing the reliability of AI algorithms in localising cephalometric landmarks on lateral cephalometric radiographs with the manual tracing of these points; differences between various algorithms were also examined [11,12,13,14,15,16,17,18,19,20,21,22,23,24,25,26,27,28,29,30,31,32,33].

### 3.1. Developing Automatic Localisation of Cephalometric Landmarks

In 2009, Leonardi et al. verified the algorithm’s reliability for the automated identification of cephalometric landmarks designed by the authors [11]. A total of 41 digital lateral cephalometric radiographs of patients aged 10–17 years were used; the patients had undergone orthodontic treatment, and different types of malocclusion were included. Ten hard tissue cephalometric landmarks were marked by five experienced clinicians in the horizontal and vertical planes. Their measurements were averaged to obtain arbitral localisation of each point. Then, the localisation of the cephalometric points was performed using the AI algorithm. The differences in the localisation of points determined using AI as opposed to the localisation of experienced clinicians, which was deemed “true”, were assessed. It was demonstrated that the differences between algorithm-located points and the mean localisation of points by the orthodontists were very small and did not exceed 0.59 mm. Statistically significant differences were shown in the horizontal plane for the following landmarks: Nasion (0.217 mm), A point (0.596 mm), B point (0.161 mm), Upper Incisor Edge (0.172 mm) and Lower Incisor Edge (0.226 mm). In the vertical plane, statistically significant differences in localisation included the following points: Nasion (0.483 mm) and Porion (0.538 mm). Tanikawa et al. in 2010 confirmed that automatic recognition of anatomic features on cephalograms is accurate and reliable also in preadolescent children with mixed dentition [12].

A fully automatic landmark annotation system (FALA), which follows a machine learning approach using Random Forest regression-voting and Constrained Local Model framework was developed by Lindner et al. [13]. The system was trained and validated using 19 landmarks located in 400 cephalograms from patients aged 7–76 years. Two experienced orthodontists independently traced manually all radiographs to obtain the “ground truth” annotations, and eight measurements were used, such as SNA, SNB, ANB and others, to evaluate skeletal malformations. The FALA system located, on average, 84.7%/96.3% of all landmarks within a 2 mm/4 mm precision range, while the manual inter-observer precision range was 62.1%/85.0%, respectively. The authors concluded that the system is very promising for conducting a fully automatic cephalometric analysis.

Park et al. compared the accuracy of the automated tracing of cephalometric points via two algorithms: You-Only Look-Once version 3 (YOLOv3) and Single Shot Multibox Detector (SSD) [14]. The accuracy of both algorithms was verified by comparing the localisation of anthropometric points against arbitrarily determined manual localisation of points by an experienced orthodontist. The study was based on 1311 cephalometric radiographs obtained from the medical dataset of radiographs (INFINITT Healthcare Co., Ltd., Seoul, Korea). Both tested algorithms were supposed to automatically determine the position of 80 landmarks. YOLOv3 surpassed SSD in accuracy for 38 out of 80 landmarks, while the remaining 42 did not show statistically significant differences between the two methods. There was no single landmark which the SSD algorithm would identify with higher accuracy, and YOLOv3 showed approximately 5% higher SDR in all ranges. The successful detection rate for the YOLOv3 algorithm was 80.4% for 2 mm ranges. The analysis of one cephalogram via the YOLOv3 algorithm lasted 0.05 s, while the SSD algorithm needed 2.89 s to perform the same analysis.

The following study was based on the same set of cephalometric radiographs and compared YOLOv3 with human examiners [15]. AI showed better accuracy in 14 out of 46 skeletal landmarks, the human examiner performed better in 14 out of 46 landmarks, and the remaining 18 out of 46 did not show statistically significant differences. For the soft tissue landmarks, the YOLOv3 showed better accuracy in 5 out of 32 landmarks, while the human examiner performed better in 7 out of 32, and the remaining 20 out of 32 did not show statistically significant differences. Interestingly, the mean difference between AI and human examiners was similar to the mean difference between human examiners (1.46 ± 2.97 millimetres versus 1.50 ± 1.48 mm, respectively). Gender, skeletal classification, image quality and the presence of metallic artefacts did not affect the AI’s accuracy in the localisation of landmarks. The YOLOv3 system was also used in a study by Moon et al. (2020) for the examination of 80 landmarks in 2400 cephalograms [16]. Their results proved that the accuracy of AI increased linearly with the increasing number of learning data sets on a logarithmic scale.

Lee et al. obtained 400 cephalometric radiographs from the ISBI 2015 Challenge dataset and assessed them using an automated framework for the detection of cephalometric points using Bayesian BCNN [17]. The characteristics of the patients included was not given. The authors compared results obtained by two junior and senior orthodontists, who had traced manually cephalometric points, with those obtained using AI. The successful detection rate was 82.11% for the 2 mm confidence interval value. The biggest discrepancy was seen for the point Soft Tissue Pogonion, while the point Sella was the easiest to identify. In conclusion, the authors stated that there was a high accuracy of the applied automated framework with the averaged values obtained manually.

The study of Kunz et al. presented the assessment of the accuracy of automated tracing of cephalometric landmarks using an AI algorithm created by the authors [18]. A total of 1792 cephalometric radiographs from a private dental office were evaluated. However, no information was given regarding the characteristics of the included patients. Twelve linear and angular measurements were assessed using 18 cephalometric points. To assess “true” values of measurements, the analysis was carried out by twelve orthodontists, and the obtained results were averaged. It was demonstrated that there was a very high correlation between AI predictions and the “gold standard” of manual localisation of points. Absolute mean differences between the two analyses were less than 0.37° for angular measurements, less than 0.20 mm for all the metric parameters and less than 0.25% for the proportional parameter of the facial height. These values did not show statistically significant differences between AI predictions and the human “gold standard”, except for the angle SN-MeGo with a *p*-value equal to 0.043, which was the only parameter with a 0.31-degree deviation.

Kim et al. examined the reliability of their own DL-based algorithm using 2075 lateral cephalometric radiographs that were taken for orthodontic purposes in two medical centres [19]. Medical records of patients, regardless of age, gender or type of malocclusion, were included in this study. A total of twenty-three points were marked on radiographs by two experienced orthodontists, and the localisation of these points was used as a reference during AI reliability verification. The landmarks included both hard tissue (facial bones and teeth) and soft tissue landmarks. Successful detection rates (SDR) of the anthropometric points were 84.3% with a 2 mm margin of error against arbitrarily marked landmark positions. The algorithm took 0.4 sec on average to recognise the localisation of all the points on a given radiograph. The mean time of manual determination of all the points by the orthodontists was not presented, nor was the interexaminer reliability of the two clinicians provided.

Kim et al. created their own programme using artificial intelligence, which was supposed to identify points on cephalometric radiographs automatically [20]. A total of 950 lateral cephalometric radiographs, taken at the Maxillofacial Surgery Clinic of the University Hospital in Yonsei in South Korea, were used to trace thirteen hard tissue points. In this study, the clinically accepted margin of error was the difference in the measurements taken by two experienced orthodontists and not the standard 2 mm margin of error adopted by other researchers. Both clinicians identified thirteen landmarks. The accuracy detection index between the two orthodontists, as assessed using AI, was 36.2% on average, and for the points Orbitale and Porion, it was 7.3% and 3.3%, respectively. Higher accuracy was observed when AI detected the following points, Nasion, A point, Menton, Upper Incisor Border, Lower Incisor Border and Anterior Nasal Spine, for which the accuracy index for the inter-examiner difference exceeded 50%. This study has shown that the deep learning model can achieve better results for some landmarks than experienced clinicians, and the inter-examiner variability is very important for assessing the effectiveness of AI detection of cephalometric landmarks.

Tanikawa et al. evaluated the clinical applicability of an automated system for the identification of cephalometric landmarks with the aim to identify errors related to patients’ factors and a minimum number of images required for the re-learning [21,22]. The authors confirmed the effectiveness of AI in various patient groups. Approximately 5–10% of the original data set of cephalograms is required for system re-training.

Yao et al. examined the reliability of their own algorithm in evaluating lateral cephalometric radiographs [23]. The study material consisted of 512 radiographs of patients from the Maxillofacial Surgery Clinic in Sichuan, China. Patients who qualified for the study aged 9–40 years, and the number of women and men was similar. Two experienced orthodontists manually traced 37 points. The inter-examiner reliability was not assessed. However, the averaged values of their measurements were treated as “true values” for the AI analysis. It was demonstrated that the accuracy of their algorithm was 97.30% for the 2 mm margin of error, and the duration of localisation of 37 anthropometric points was 3 s. The pronosale point had the highest accuracy value (SDR for 2 mm = 99%), while the Pogonion point had the smallest accuracy, for which SDR was 76% for the 2 mm margin of error.

Another study by Uğurlu assessed the accuracy of the automated detection of cephalometric points via the authors-designed AI algorithm [24]. The study material consisted of 1620 radiographs of patients aged 9–20 years treated at the Orthodontic Clinic of the Eskisehir Osmangazi University in Turkey. An experienced orthodontist manually identified 21 hard and soft tissue cephalometric points, which constituted reference localisations for testing the accuracy of the automated determination of points via the AI algorithm. The value of the SDR index was, on average, 76.2% for the 2 mm error value. The algorithm’s accuracy for the Sella, Nasion, Orbitale, A point and B point was 98.3%, 77.8%, 66.1%, 76.1% and 66.1%, respectively, for the 2 mm measurement error. The lowest accuracy was for the point Gonion, with 48.3% for the 2 mm error. Recently, Popova et al. confirmed that the presence of orthodontic appliances did not significantly influence the performance of CNN-based open-source models, such as the Python programming language [25].

### 3.2. Commercial Software/Applications

Jeon et al. compared the outcomes of the conventional cephalometric analysis with the commercially available software CephX based on lateral head radiographs from 35 adult individuals [26]. Significant differences were found in the localisation of saddle angle, linear measurements of maxillary incisors to the NA line and mandibular incisors to the NB line. There were no significant differences in the localisation of the two soft tissue landmarks. The authors stated that the widths of limits of agreement were wider for the dental measurements compared to the skeletal measurements and concluded that automatic cephalometric analyses based on CNN might offer clinically acceptable clinical performance.

An automated cephalometric analyser Ceppro was tested on 110 cephalometric radiographs for the detection of 16 cephalometric landmarks (Bulatova et al., 2021) [27]. The software was initially trained on 15,000 cephalograms with a 1:1 scale obtained from one cephalometric machine at the Seoul National University Dental Hospital (Seoul, Korea).

AI errors were marked in 38 images out of 110. The authors concluded that the tested system facilitates cephalometric analysis in daily clinical practice and the assessment of bigger databases for research purposes. However, different artefacts may affect its effectiveness.

Ristau et al. examined the accuracy of automated detection of landmarks on lateral cephalometric radiographs using the AudaxCeph commercial software that utilises AI [28]. The study was based on 60 archived radiographs of patients presenting for orthodontic treatment at the University of Louisiana, USA. The inclusion criteria included the presence of all permanent teeth except third molars. On each radiograph, two experienced orthodontists independently marked 13 anthropometric points. The same points were later automatically marked using AudaxCeph. The inter-examiner reliability between the two orthodontists and between each orthodontist and the AI software was assessed. It was demonstrated that the differences in the mean positions of the points marked by both orthodontists did not exceed 2 mm, which is the clinically accepted margin of error during cephalometric analysis. Likewise, the positions marked using AudaxCeph did not differ from the measurements obtained by any of the two practitioners by more than 2 mm, except for the points Porion and Lower Incisor Apex, where the algorithm deviated more than 2 mm in the horizontal or vertical planes.

Kılınçi et al. assessed differences between values from the cephalometric analysis obtained in three different ways [29]. Hand-tracing cephalometric analysis performed by the orthodontist was compared with AI (WebCeph software) and the CephNinja (version 4.2) smartphone application, which involves the manual identification of landmarks by an orthodontist on the screen of their phone. Magnifying and decreasing the size of the picture was applied to pinpoint the areas. The study material consisted of 110 cephalometric radiographs from the archives of the orthodontic clinic of Aydin University in Istambul, Turkey. The patients were in the age range of 10–24 years. Each of the three methods was used to perform cephalometric analysis and to obtain values for 11 linear and angular measurements. SNA, SNB, SN-MP angle, U1-SN angle, L1-NB (mm) and E Line-Upper Lip values differed considerably for each method, and these differences were clinically significant. The authors demonstrated that the accuracies of WebCeph and CephNinja software were markedly worse than the hand tracing of landmarks, thus restricting their clinical applicability.

Çoban et al. examined 105 cephalometric radiographs of patients who presented at the Department of Orthodontics at Erciyes University in Turkey to undergo orthodontic treatment [30]. The inclusion criteria included patients older than 15 years. The radiographs were subjected to cephalometric analysis, including 23 measurements obtained with two methods: manual tracing of cephalometric points by an orthodontist and automated detection using AI-based software. It was shown that the results obtained with the WebCeph software were statistically significantly different in comparison with the standard cephalometric method in which the orthodontist was responsible for the tracing. Major differences were detected for the SNA, ANB, NA, Y-axis and SN. GoGn, SN.PP, ANS-Me, CoA, CoGn, U1.PP, U1-NA, IMPA, L1-NB, L1.NB, NLA and ULE measurements. It was also demonstrated that 17 out of 23 comparable measurements showed discrepancies between the two applied methods of cephalometric analysis.

Also, Mahto et al. compared the values from the cephalometric analysis performed manually with the AI-based, WebCeph software [31]. A total of 18 landmarks and 12 angular and linear measurements were analysed on 30 cephalometric radiographs obtained from the Department of Orthodontics of the Dhulikhel Hospital in Nepal before orthodontic treatment. The mean age of patients was 20.17 years ± 6.72. Manual cephalometric analysis was performed by one clinician, and the results were compared with the values provided by the WebCeph. The intraclass correlation coefficient (ICC) indicated a high correlation between the values obtained using both methods, which was demonstrated in 7 out of 12 examined measurements. They included ANB, FMA, IMPA, LL to E-line, L1 to NB (mm), L1 to NB (°) and S-N to Go-Gn. For the remaining five measurements, the ICC was between 0.75 and 0.9. The small number of examined cephalometric radiographs was a major limitation of this study.

Tsolakis et al. performed the cephalometric analysis of 100 radiographs of patients who presented for orthodontic treatment at a private dental office [32]. The medical records of patients included in the study did not account for gender or age. The accuracy of automated detection of points and measurements taken using CS imaging V8 was compared with manual tracing conducted by an experienced orthodontist. Mean values of the SNA, SNB, ANB, SN-MP, U1-SN, L1-NB, SNPg, ANPg, SN/ANS-PNS, SN/GoGn, U1/ANS-PNS, L1-APg, U1-NA and L1-GoGn measurements did not reveal statistically significant differences between the two methods. Some measurements, however, did demonstrate larger discrepancies between the automated and manual determinations of points. Significant differences were observed for FMA measurements (2.1 degrees), IMPA (4 degrees), ANS-PNS/GoGn (3 degrees) and U1-L1 (3.3 degrees).

Recently, Jiang et al. developed a novel and accurate system for automatic cephalometric landmark location and analysis based on a two-stage cascade CephNet system [33]. The system consisted of two-stage neural networks, from which the first aimed to detect 10 regions of interest (ROI), each containing 1–9 landmarks, and in the second stage, the landmarks were accurately located in the ROIs. The system was not dependent on the cephalography machine or anatomical variability of patients, and 9870 cephalograms from 20 medical institutions in China were included for evaluation and training. Five orthodontists manually annotated 30 skeletal, dental and soft tissue cephalometric landmarks to establish the standard data for comparison and training of AI, and in the next stage, 100 orthodontists adjusted the landmark location using web browsers (SaaS system). The accuracy of automatic landmark localisation reached, on average, 66,15% within 1 mm SDR and 91.73% within 2 mm. The least accurate landmark location was seen for the point Gonion (Go), and landmarks surrounding stable and clear anatomical structures exhibited higher SDR than landmarks interfered with by overlapping anatomical structures. The landmarks defined as the “most front” or “most convex”, for example, Pog) have also shown larger detection errors. The accuracy of cephalometric analysis based on 11 cephalometric measurements was, on average, 89.33% for CephNet.

### 3.3. Successful Detection Rates (SDR)

Hwang et al. compared the accuracy of the YOLOv3 algorithm with the accuracy of manual identification of anthropometric points by two orthodontists: the first had 28 years of clinical experience, while the other was a 3rd-year orthodontic resident [15]. Both were employed at the same institution. The study material was the same as in part 1 of their research [14]. It was demonstrated that the YOLOv3 algorithm detected the same position for each landmark, while the human inter-examiner variability of manual detections demonstrated a detection error of 0.97 ± 1.03 mm. The mean difference between human examiners was 1.50 ± 1.48 mm. Comparisons in the detection errors between AI and human examiners for the identification of all cephalometric points were less than 0.9 mm, except for one landmark at the apex of the lower incisor, where measurement showed an error of 1.2 mm. The authors concluded that the differences between AI algorithms and human examiners did not seem to be clinically significant.

A similar study was conducted by Bulatova et al. in which the accuracy of the automated determination of points via the YOLOv3 algorithm was compared with manual tracing performed by one orthodontist [27]. Cephalometric analysis was conducted on 110 cephalographs obtained from the AAOF Legacy Denver medical database. Sixteen points were identified with both methods. The successful detection rate of AI with an accuracy of up to 2 mm when manual detection and AI were compared was 75% and 93% with an accuracy of up to 4 mm, which the authors believe to be clinically acceptable. The points for which the localisation difference was greater than 2 mm were L1 apex, U1 apex, Basion, Gonion and Orbitale.

### 3.4. Size of the Dataset and Patients’ Characteristics

Tanikawa et al. were the first to evaluate the effectiveness of the automatic system for recognition of cephalometric landmarks in patients with full permanent dentition (400 patients, mean age: 23.6 years) and patients with mixed dentition (459 patients, mean age: 8.9 years) [12]. The authors concluded that the system successfully recognised all the anatomic structures surrounding all the landmarks. The mean success rate was 84%, with a range from 60% (N landmark) to 100% (Ptm landmark).

Moon et al. determined the size of the dataset for AI algorithm training to make it as reliable as a human hand that manually performs cephalometric analysis [16]. It was estimated that the mean difference in a manual tracing of cephalometric landmarks by different orthodontists was 1.5 mm. The study assessed the accuracy of AI cephalometric analysis by changing the number of cephalograms available for the AI training from 50 to 2000 radiographs and the number of marked points in one picture from 19 to 80. It was shown that the accuracy of AI increased linearly in proportion to the increasing number of learning data sets (cephalometric radiographs) on a logarithmic scale. It decreased with the increasing number of detection targets (points needed for a specific cephalometric analysis). It was calculated that at least 2300 sets of learning data were necessary to make the algorithm as accurate as human examiners are.

Tenikawa et al. examined the accuracy of an AI algorithm, which was used for the automated detection of cephalometric landmarks depending on patients’ characteristics such as dental age, the use of orthodontic devices, presence of clefts lip and/or palate and overjet [21]. For training and testing the algorithm, a total of 1785 cephalometric radiographs of patients who presented for orthodontic treatment at the University Dental Hospital in Osaka, Japan, were used. The patient’s age ranged from 5.4 to 56.5 years. It was demonstrated that the successful detection rates (SDR) differed in subgroups from 85% to 91%. It was also observed that cleft of the lip and/or palate was a factor responsible for higher identification errors, while the dentition type, orthodontic devices and the extent of overjet were not significant factors.

Artificial intelligence is used for copying previous solutions in trained samples. Therefore, it cannot guarantee exact solutions if the image has new input data (picture quality) or if the picture considerably differs from the samples used for training. In order to overcome this problem, AI should re-learn additional samples that are similar to the new data input. For example, a system that was trained to be used in adult patients and has shown 88% accuracy in adults may still drop to 69% when used in children, even if the radiation dose was maintained. After the system was re-trained using 400 additional cephalograms of children, the success rate increased to 82% [12]. In comparison, to obtain high accuracy of landmark identification for cephalometric analysis of radiographs taken with a different cephalograph (with a different quality), the AI algorithm has to be additionally trained with 85–170 radiographs having a specific desired quality, which amounts to approximately 5–10% increase in the initial data set [22].

Recently, Popova et al. assessed the influence of growth structures such as tooth buds and the presence of fixed orthodontic appliances on the accuracy of a customised CNN model for the automatic detection of cephalometric landmarks [25]. Sixteen skeletal and dental cephalometric landmarks were included in the analysis. Two last year orthodontic residents and an orthodontic specialist created a verified dataset, which was used as a reference for the training, testing and validation of the CNN model. In total, 890 cephalograms were used, from which the training dataset consisted of 430 cephalograms with both mixed and permanent dentition and orthodontic appliances. The performance of the developed CNN was tested using 460 cephalograms with various radiographic features, such as fixed orthodontic appliances and anatomical structures in patients at different growth stages. Significant differences were observed in the recognition of the Ap-Inferior point and the Is-Superior points between patients with mixed and permanent dentition. Fixed orthodontic appliances, such as brackets, bands, and other fixed orthodontic appliances, had no significant effect on the performance of the CNN model. The growth structures, such as tooth germs in mixed dentition, play a role in the performance of the AI model.

## 4. Discussion

The majority of the studies included in this review have been published in the last three years, which shows a rapid increase in interest in the application of AI in cephalometric analysis and orthodontic diagnostics of malocclusions. The accuracy of different types of AI algorithms varies, as demonstrated by the results published in the included studies (Table 1). The authors used different numbers of cephalograms for the testing and validation of the database, which varied from dozen to a thousand. Also, the number of clinicians performing the manual annotation of landmarks varied in number and in clinical experience in cephalometric tracing. Moon et al. (2020) concluded that the more data that were implemented during the training procedure of AI, the smaller the detection errors observed [16]. The development of reliable “gold standards” in the identification of cephalometric landmarks is important to reduce bias in the dataset used for AI training. Also, the time of the AI analysis varied between studies.

Today, CNN-based algorithms derived by many authors for the purpose of their studies, or YOLOv3 or SSD algorithms, not available to doctors in their daily clinical practice, are more effective and accurate than the widely available web-based software such as WebCeph, AudaxCeph or CS Imaging.

Most AI algorithms used for the automated tracing of landmarks on lateral cephalographic radiographs are characterised by relatively high accuracy. In most studies, the confidence interval was within 2 mm, and the mean percentage of detected landmarks within this margin was above 80%. However, from the clinical point of view, the localisation error up to 2 mm can be acceptable for some, but not all points traced in cephalometric analysis. The localisation of cephalometric points A and B in the horizontal plane is crucial for the determination of maxillary/mandibular relations in the sagittal plane. An inaccurate localisation of these points in the range of 1.5–2 mm would result in a considerable inaccuracy of many angular and linear measurements, especially if errors are duplicated using the same landmark in several measurements. It also has to be stressed that cephalometric analysis of lateral head radiographs performed manually is a subjective examination, and the localisation of specific anthropometric points may differ between orthodontists. It has been demonstrated that the mean discrepancies between two experienced clinicians could be up to 1.5 mm as well. Moreover, a repeated tracing of landmarks on the same radiograph by one orthodontist may entail an error of approximately 1 mm between two measurements. Unlike manual tracing of cephalometric landmarks, the AI algorithm always marks identical localisation of the landmarks, which can be an additional asset for its use [15].

The studies confirmed that the time needed for analysing a cephalometric radiograph using most of the popular AI algorithms takes a few seconds. This is considerably shorter than the manual tracing of landmarks by clinicians. The most recent algorithms evolve rapidly, and their calculating capacity increases, which will probably result in their increased efficiency and reliability. It can be expected that in the future, AI algorithms that are used for the automated localisation of landmarks may be more accurate than manual tracing. At the same time, the interpretation of cephalometric analysis via artificial intelligence may be inferior to the interpretation performed by experienced orthodontists but can still be useful to less experienced specialists or even non-specialists. It is necessary to conduct further studies to assess the reliability of AI-performed cephalometric analysis in planning, monitoring and analysing orthodontic treatment. There is no doubt that the ease and short duration of cephalometric analysis via AI may be a significant factor in facilitating orthodontic treatment in clinical practice.

The use of AI algorithms in radiological diagnostics in the area of orthodontics is not restricted to the automated detection of landmarks in cephalometric analysis. AI provides high accuracy in the assessment of cervical vertebral maturation on radiographs [34,35]. Another AI algorithm that is described in the literature is supposed to predict the need for tooth extractions due to orthodontic reasons [36].

The identification of cephalometric landmarks is challenging, as a skull is a 3D object projected onto a 2D plane on a lateral head cephalogram. Overlapping structures increase the difficulty in precise landmark identification, especially in patients with facial asymmetry. Moreover, improper head position during image acquisition and radiographic distortions may lead to errors in landmark identification by orthodontic professionals. The quality of cephalograms used for landmark identification, the level of orthodontic training and experience in landmark identification as well as inter-observer variability between clinicians who participate in the training and validation of the AI model are important factors and limitations of this diagnostic tool. Another source of AI inaccuracy might be due to the operator’s mistake while calibrating images for the AI cephalometric analysis, like in the Ceppro software (Bulatova et al., 2021) [27]. Even a small error in using a digital ruler alters the number of pixels in 1 mm and can influence the coordinates for all points.

The advantage when using an automated system for the identification of cephalometric landmarks in comparison with the manual annotation is the fact that it would always give the same result for the same image, while there are large variations in the accuracy of manual annotation related to the levels of training and experience [13]. Improving the training and validation of AI algorithms may completely replace manual cephalometric tracing in the future.

Threats and challenges of the future use and development of AI in the analysis of patients’ medical records are related to the data protection and application of the principles of medical ethics whenever computer software that simulates human brain activity is used. It is possible that new legal regulations concerning the application of AI in the diagnostics and monitoring of orthodontic treatment will have to be proposed and implemented. Pre- and postgraduate curricula and clinical practice must be adjusted for technological advancements, so they can contribute to the optimisation of orthodontic treatment without adversely affecting its effectiveness.

## 5. Conclusions

In recent years, artificial intelligence has been more and more frequently used in the orthodontic diagnostic process. It is a promising tool that facilitates the tracing of cephalometric landmarks in daily clinical practice, which can assist less experienced clinicians in orthodontic treatment planning and shortens the time devoted to performing radiological diagnoses of patients. At the same time, the reliability of AI in cephalometry differs depending on the accuracy of manual landmark identification related to the operator’s clinical training and experience, the number and quality of radiographs, the type of algorithm or application, which have to be accounted for during interpretation of results. It is predicted that AI will continue to be implemented and further developed for its application in orthodontics.

## Figures and Tables

**Table 1 diagnostics-13-02640-t001:** Studies on the effectiveness of AI in the analysis of lateral cephalometric radiographs.

No	Study	No. of Cephalograms	Patients’ Age (in Years)	Type of Algorithm	No. of Examiners	No. of Landmarks/Mean SDR	No. of Measurements/Mean Error	Time for Analysis (in Seconds)
1	Leonardi et al., 2009 [11]	41	10–17	Authors’ algorithm/CNN, Borland C++	5	10/ n.s.	n.s.	257 for 10 landmarks
2	Tanikawa et al., 2010 [12]	859 (400: permanent dentition; 459: mixed dentition)	5–60; mean age: 23.6 (permanent dentition); 8.9 (mixed dentition)	Authors’ algorithm/PPED system	2	18/ n.s.	n.s.	n.s.
3	Lindner et al., 2016 [13]	400	7–76	Authors’ algorithm/FALA system, RFRV-CLM	2	19/ 84.7% in the range of 2 mm	8/ 78.4 ± 2.61%	<3
4	Park et al., 2019 [14]	1311 (1028: training set; 283: testing set)	n.s.	Authors’ algorithm/YOLOv3 and SSD	1	80/ YOLOv3: 80.4% in the range of 2 mm	n.s.	0,05 for YOLOv3; 2.89 for SSD
5	Hwang et al., 2020 [15]	1311 (1028: training set; 283: testing set)	n.s.	Authors’ algorithm/YOLOv3 and manual analysis	2	80/ mean detection error: 1.46 ± 2.97 mm	n.s.	n.s.
6	Moon et al., 2020 [16]	2400 (2200: training set; 200 test set)	n.s.	Authors’ algorithm/YOLO v3	2	80/ n.s.	n.s.	n.s.
7	Lee et al., 2020 [17]	400	n.s.	Authors’ algorithm/Bayesian CNN	2	19/ 82.11% in the range of 2 mm	n.s.	512/38 for 19 landmarks (1 GPU/4 GPU)
8	Kunz et al., 2020 [18]	1792 (96.6%: training set; 3.4% validation set)	n.s.	Authors’ algorithm/CNN, Keras and Google Tensorflow	12	18/ n.s.	12/ <0.37° (angular measurements); <0.20 mm (metric measurements); <0.25% (proportional measurements)	n.s.
9	Kim at al., 2020 [19]	2075	n.s.	Authors’ algorithm/DL, SHG, Tensorflow, Python	2	23/ 84.7% in the range of 2 mm	n.s.	0.4 for 23 landmarks
10	Kim et al., 2021 [20]	950 (800: training set; 100: validation set; 50: testing set	n.s.	Authors’ algorithm/CNN	2	13/ 64.3% in the range of 2 mm	n.s.	n.s.
11	Tanikawa et al., [21]	1785	5.4–56.5; mean age: 12.2	Authors’ algorithm/CNN-PC & CNN-PE, Adam	2	26/ success rates from 85% to 91%	n.s.	n.s.
12	Tanikawa et al., 2021 [22]	2385	5.8–77.9	Authors’ algorithm/ CNN-PC&PE, Adam	2	26/ success rates from 85% to 90%	n.s.	n.s.
13	Yao et al., 2022 [23]	512 (312: training set; 100: validation set; 100: testing set)	9–40	Authors’ algorithm/CNN, PyTorch	2	37/ 45.95% in the range of 1 mm; 97.3% in the range of 2 mm	n.s.	3 for 37 landmarks
14	Uğurlu, 2022 [24]	1620 (1360: training set; 140: validation set; 180: testing set)	9–20	Authors’ algorithm/CNN/PyTorch, Python	1	21/ 76.2% in the range of 2 mm	n.s.	n.s
15	Popova et al., 2023 [25]	890 (387: training set; 43: validation set; 460: testing set)	All ages	Authors’ algorithm/CNN/(Keras and TensorFlow, Python	3	16/ 84.73% in the range of 2 mm	n.s.	n.s.
16	Jeon et al., 2021 [26]	35	Mean age: 23.8	Commercial analysis/CephX	1	16	26/ 0.1–0.3° (angular measurements); 0.1–0.3% (linear measurements)	n.s.
17	Bulatova et al., 2021 [27]	110	n.s.	Commercial analysis/Ceppro	2	16/ ±0.13 mm in the range of 2 mm for 75% of landmarks; mean difference 2.0 ± 3.0 in X plane and 2.1 ± 3.0 in Y plane	n.s.	n.s.
18	Ristau et al., 2022 [28]	60	Patients with a full complement of teeth	Commercial analysis/AudaxCeph	2	13/max. mean error: <2.6 mm in X plane; <2.3 mm in Y plane	n.s.	n.s.
19	Kılınç et al., 2022 [29]	110	10–24, mean age: 15.83 ± 2.85	Commercial analysis/ WebCeph and CephNinja	1	n.s.	11/ ICC from 0.170 to 0.884	n.s.
20	Çoban et al., 2022 [30]	105	>15, mean age: 17.25 ± 2.85	Commercianalyser/ WebCeph	1	n.s.	22/ ICC from 0.418 to 0.959	n.s.
21	Mahto et al., 2022 [31]	30	Mean age: 20.17 ± 6.72	Commercianalyser/WebCeph	1	n.s.	12/ ICCC from 0.795 to 0.966	n.s.
22	Tsolakis et al., 2022 [32]	100	Mean age: 15.9 ± 4.8	Commercial analyser/CS imaging V8	1	16	18/ ICC from 0.70 to 0.92	n.s.
23	Jiang et al., 2023 [33]	9870 (8611: training set; 1000: validation set; 259: testing set)	6–50	Commercial analyser/CNN/CephNet	5/100	28/ 66.15% in the range of 1 mm; 91.73% in the range of 2 mm	11/ 89.33%	n.s.

CNN: Convolutional Neural Network; CNN-PE: Convolutional Neural Network for Point Estimation; CNN-PC: Convolutional Neural Network for Patch Classification; DL: Deep Learning; GPU: Graphic Processing Unit; ICC: Inter-Method Correlation Coefficient; n.s.: Not stated; PPED: Projected Principal Edge Distribution; RFRV-CLM: Random Forest Regression Voting-Constrained Local Model; SHG: Stacked Hourglass Network.

## Data Availability

The data presented in this study are available upon request from the corresponding author.

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
