# Peer review of "Application of Artificial Intelligence (AI) in a Cephalometric Analysis: A Narrative Review"

_diagnostics, 2023, doi:10.3390/diagnostics13162640_

Round 1
Reviewer 1 Report
The reviewer feel the topic is quite interesting and may draw reading interest from the scientific community. However, the contents of this review need to be improved.
1, The article focuses on the application of AI in cephalometric analysis. A revision of the title is suggested.
2, Please explain the inclusion and exclusion criteria for the literature search and the reason why the author only searched two databases, as some other relevant literature was not been included in the review, such as PMID: 27645567.
3, Several review articles explored the artificial intelligence used in cephalometric analysis studies (PMID: 36604364, PMID: 36553978, PMID: 35757486, PMID: 34046742). How this review is different from the previously published reviews?
4, The authors are encouraged to make a discussion among different types of algorithm models in the different studies, and provide more of their own perspective in how to better improve the algorithm to further improve the accuracy and reliability.
5, It is recommended to list the sample size, algorithm type, the number of examiner, number of landmark points, interexaminer reliability, analysis time, successful detection rates and other parameters for each study in tabular form, in order to allow for clearer and more intuitive comparison.
Author Response
Many thanks for your time to review our paper and for your valuable comments to the manuscript.
We have revised the manuscript according to Your comments as below:
- The title was changed to: Application of artificial intelligence (AI) in a cephalometric analysis: a narrative review.
- We have extended our search to the Scopus and Dentistry & Oral Sciences Source databases and included the article PMID: 27645567 as suggested. After the extended search
- The other articles on the application of AI in cephalometrics did not included the latest’s articles and used other databases. In particular: 1. PMID 34046742: Chaurasia A, Arsiwala L, Lee JH, Elhennawy K, Jost-Brinkmann PG, Demarco F, Krois J. Deep learning for cephalometric landmark detection: systematic review and meta-analysis. Clin Oral Investig. 2021 Jul;25(7):4299-4309. The search included papers published in the years 2017-2020, also other databases were used - (Medline/Embase/IEEE/arXiv). 2. PMID 35757486: Subramanian AK, Chen Y, Almalki A, Sivamurthy G, Kafle D. Cephalometric Analysis in Orthodontics Using Artificial Intelligence-A Comprehensive Review. Biomed Res Int. 2022 Jun 16;2022:1880113. The search did not included articles published after August 2021, also other databases were used. (Google Scholar, EMBASE, and Science Direct). 3. PMID 36553978: Junaid N, Khan N, Ahmed N, Abbasi MS, Das G, Maqsood A, Ahmed AR, Marya A, Alam MK, Heboyan A. Development, Application, and Performance of Artificial Intelligence in Cephalometric Landmark Identification and Diagnosis: A Systematic Review. Healthcare (Basel). 2022 Dec 5;10(12):2454. The search did not included articles published after October 2021. 4. PMID 36604364: de Queiroz Tavares Borges Mesquita, G., Vieira, W.A., Vidigal, M.T.C. et al. Artificial Intelligence for Detecting Cephalometric Landmarks: A Systematic Review and Meta-analysis. J Digit Imaging 36, 1158–1179 (2023). The search did not included articles published after November 2021, the article was not published at the time of the article’s search and preparation of the the current manuscript.
- We have included the comments on the limitations AI in cephalometric analysis especially related to the baseline information used for training the algorithm and factors of importance for their use in clinical practice, for example dental age.
- We have included a table summarizing our search according to your suggestions. We hope, that it will address your recommendations.
Reviewer 2 Report
Actually, it is a very good review. It really shows the importance of the extra efforts that should be exerted by researchers to improve the quality of automatic detection of radiographic landmarks.
Author Response
Dear Reviewer,
Many thanks for your time to review our paper and for your positive response to our paper.
Reviewer 3 Report
The aim of this narrative review was to present data from the literature on the effectiveness of AI in orthodontic diagnostics based on the analysis of lateral cephalometric radiographs. The study is well organized and provides valuable information about the use of Ai in rtg-cephalometric tracing. I think that the paper can be accepted in the present form.
Author Response
Dear Reviewer,
Many thanks for your time to review our paper and for a positive response.
Reviewer 4 Report
The comments are attached.

Author Response
Dear Reviewer,
Many thanks for your time to review our paper and for your valuable comments to the manuscript.
We have revised the manuscript extensively (changes in red) and therefore, we hope that the revised version will address your comments.
Below, are our replies to your specific comments
- We have aimed to provide information on the accuracy of specific cephalometric landmarks and measurements when applying AI in cephalometrics. The authors have used different datasets for training and validation of AI, different systems or applications and different aims for their analysis. We have described their outcomes as precisely as possible, also in relation to specific cephalometric landmarks or measurements when applicable, but it impossible in our opinion to provide a reliable comparison between studies regarding specific parameters and measurements at present. We have added a comment in the discussion in relation to this problem.
- We have extended our search to the Scopus and Dentistry & Oral Sciences Source databases and included 4 more articles. The included review (Alessandri-Bonetti at al., 2023) was published not published at the time of the article’s search and preparation of the current manuscript.
- We have revised the results and discussion and added a table summarizing the search. We hope, that it would address your comments.
- We are very grateful for your suggestion to use SANRA. It would be of a great value for our next narrative review, unfortunately this review was performed not using the SANRA, but we have tried to address as much as possible of the SANRA’s recommendations in the revised version of the manuscript.
Reviewer 5 Report
Dear Editor, Dear Authors, this manuscript is a great new perspective on orthodontic diagnostics.
The following changes in the conclusion paragraph should be added: how AI could be improved for better use in orthodontics (e.g. tracing, landmark detection, quality of x-rays, etc.)
The title should be adjusted according to the review methodology, as only cephalometric analysis was used, but the title is generally diagnostic and partially misleading.
Author Response
Dear Reviewer,
Many thanks for your time to review our paper and for your valuable comments to the manuscript.
We have revised the manuscript extensively adding 2 more databases ( Scopus and Dentistry & Oral Sciences) and addressed the issues raised below related the effectiveness of the AIin the main text and shortly in the conclusions. All changes included in the revised version are marked in red.
In response to your suggestions, we have changed the title of the manuscript to "Application of artificial intelligence (AI) in a cephalometric analysis: a narrative review."
Round 2
Reviewer 1 Report
The author's modifications are acceptable